# Hydrogel Based on Cashew Gum and Polyacrylamide as a Potential Water Supplier in Mombaça Grass Pastures: A Sustainable Alternative for Agriculture

Dhiéssica Morgana Alves Barros [1], Ricardo Loiola Edvan [1,*], João Paulo Matos Pessoa [1], Romilda Rodrigues do Nascimento [2], Luan Felipe Reis Camboim [2], Leilson Rocha Bezerra [2], Marcos Jácome de Araújo [1], Heldeney Rodrigues de Sousa [3] and Edson C. Silva-Filho [3]

[1] Animal Science Department, Federal University of Piauí, Bom Jesus and Teresina, Teresina 64900-000, Brazil; morganabarros1@ufpi.edu.br (D.M.A.B.); jpmatospessoa@gmail.com (J.P.M.P.); jacome@ufpi.edu.br (M.J.d.A.)
[2] Health and Agricultural Technology Center, Federal University of Campina Grande, Patos 58708-110, Brazil; romildarn01@ufpi.edu.br (R.R.d.N.); luanzootec@ufpi.edu.br (L.F.R.C.); leilson.bezerra@ufcg.edu.br (L.R.B.)
[3] Chemistry Department, Federal University of Piauí, Ministro Petrônio Portela Campus, Teresina 64049-550, Brazil; heldeney.rodrigues@ufpi.edu.br (H.R.d.S.); edsonfilho@ufpi.edu.br (E.C.S.-F.)
* Correspondence: edvan@ufpi.edu.br

**Abstract:** Hydrogels are water-absorbing polymers that can hydrate forage plants in the soil. The objective was to evaluate the replacement of synthetic hydrogels derived from petroleum with biodegradable hydrogels in Mombaça grass pastures (*Megathyrsus maximum*). The experimental treatments consisted of no hydrogel (NH); synthetic commercial hydrogel (CH), made from a synthetic polyacrylamide product; and biodegradable test hydrogel (TH), obtained from cashew gum (*Anacardium occidentale*). The experimental design consisted of randomized blocks with five replications and three treatments. The morphogenesis, production, chemical, and mineral composition of the Mombaça grass pasture were assessed. The data were subjected to analysis of variance and mean comparison using the Scott–Knott test at 5% probability. The leaf elongation rate showed 42.3 mm day$^{-1}$ in the treatment TH, which was higher ($p < 0.05$) than NH (35.0 mm day$^{-1}$). The green leaf mass yield was higher in TH than in NH and CH. On the other hand, hydration had no effect on the chemical composition. The mineral composition of Mombaça grass showed more Zn when TH was used. It can be concluded that biodegradable hydrogels can replace synthetic commercial hydrogels in pastures.

**Keywords:** *Megathyrsus maximus*; morphogenesis; organic hydrogel; planting gel; polymers; synthetic hydrogel

## 1. Introduction

Mombaça grass (*Megathyrsus maximus* cv. Mombaça) is a perennial forage plant that grows in both cespitose and clumpy ways. It is very productive when grown on fertile soils but does not tolerate waterlogging or prolonged periods of water deficit [1]. Prolonged water deficit is very damaging to the grass, as it minimizes leaf emergence and tissue expansion by the forage plant. It can also reduce tillering and affect nutrient absorption by the plant, even if the nutrient is available in adequate quantities in the soil. As a result, prolonged water deficit reduces grass productivity [2].

One strategy to reduce the effects of prolonged water deficit in the soil is to use products based on synthetic polymers derived from petroleum in solid powder form, which are water absorbers, being previously hydrated and then incorporated into the soil. In this case, they are known as hydrogels. But, in some situations, this synthetic powder is incorporated directly into the soil, in which case it is called a planting gel. When water is available, these commercial synthetic polymers can absorb more than 400 times their weight in water, releasing it slowly and gradually as the soil's humidity decreases [3].

Bio-hydrogels have emerged as a sustainable alternative to reduce the problems of non-biodegradability of synthetic polymers [4,5]. Because they are natural compounds, most biopolymers are compatible with biological systems and have low toxicity without altering their mechanical properties, which further increases the technological relevance of these materials. In a study, the natural hydrogel produced from Babaçu provided better growth and chemical composition of cactus pear genotypes [6].

The cashew tree (*Anacardium occidentale* L.) is a native fruit tree extensively cultivated in Brazil and has great economic potential. Its nuts have high commercial value and present medicinal properties [7,8]. Hydrogels derived from the chemical modification of polysaccharides, such as cashew gum (CG) with poly acrylic acid (acrylic acid), are particularly important because they have advantages in terms of ease of handling and chemical modification, as well as high production potential with characteristics similar to or better than those of 100% synthetic materials [9].

The hypothesis tested that using a natural hydrogel based on cashew gum in the pasture will result in chemical composition, morphogenic characteristics, and productivity of the Mombaça grass that is similar to or better than the synthetic commercial hydrogel. The objective of this study was, therefore, to determine the chemical composition, mineral composition, morphogenic characteristics, and productivity of Mombaça grass subjected to different hydration management with hydrogels in the pasture, as well as to evaluate the possibility of replacing the synthetic commercial hydrogel with a biodegradable one.

## 2. Material and Methods

### 2.1. Experimental Location

The study was conducted at the Cinobelina Elvas campus (CPCE) of the Federal University of Piauí (UFPI), in Bom Jesus, Piauí, Brazil, which is located at geographical coordinates of a $09°04'28''$ south latitude, a $44°21'31''$ west longitude and an average altitude of 277 m. The experiment lasted from July 2021 to July 2022, recording average precipitation and temperature values of 700 mm and 38 $°C$, respectively.

The climate of the region is classified as Aw (tropical, hot, and humid, with dry seasons from spring to summer and rainy seasons from fall to winter) according to the Köppen classification. The prevalent vegetation in the region is the Cerrado, which is equivalent to the Brazilian savannah [10].

### 2.2. Experimental Treatments

Treatments consisted of three hydration methods for Mombaça grass pasture: no hydrogel (NH), in which no hydrogel was added; a test hydrogel (TH), which is biodegradable based on Cashew gum; and a commercial hydrogel (CH), which is based on polyacrylamide. TH and CH were applied in the amount of 20 kg ha$^{-1}$ to the Mombaça grass pasture. A randomized block design was adopted with five replications and three treatments.

### 2.3. Hydrogels Used

The biodegradable test hydrogel obtained from cashew gum was produced using a copolymerization process with added nutrients (Figure 1). A 2.0 g sample of cashew gum was dispersed in 30.0 mL of water under agitation and a nitrogen atmosphere to reduce the inhibitory effect of oxygen on the radical polymerization reaction. Next, 0.024 g of potassium phosphate and 2.10 g of an acrylamide monomer were added to the supernatant, followed by 0.016 g of a KPS initiator, 0.024 g of NN-methylenebisacrylamide, and 100 mg of potassium bicarbonate. After 5 min of stirring and bubbling with nitrogen, 100.0 L of the TEMED accelerator was added. The system was closed and kept under nitrogen and stirring until the gel point was reached. Once obtained, the gel was washed in a 30% methanol/water solution to remove the acrylamide homopolymer and then freeze-dried to a solid mass. The synthesized gels (1.0 g) were subjected to an alkaline hydrolysis reaction with 40.0 mL of NaOH (0.5 mol/L) to convert their amide groups into carboxylate, then

washed and freeze-dried. All the reagents used were obtained from Aldrich (São Paulo, Brazil), in analytical grade, without prior treatment.

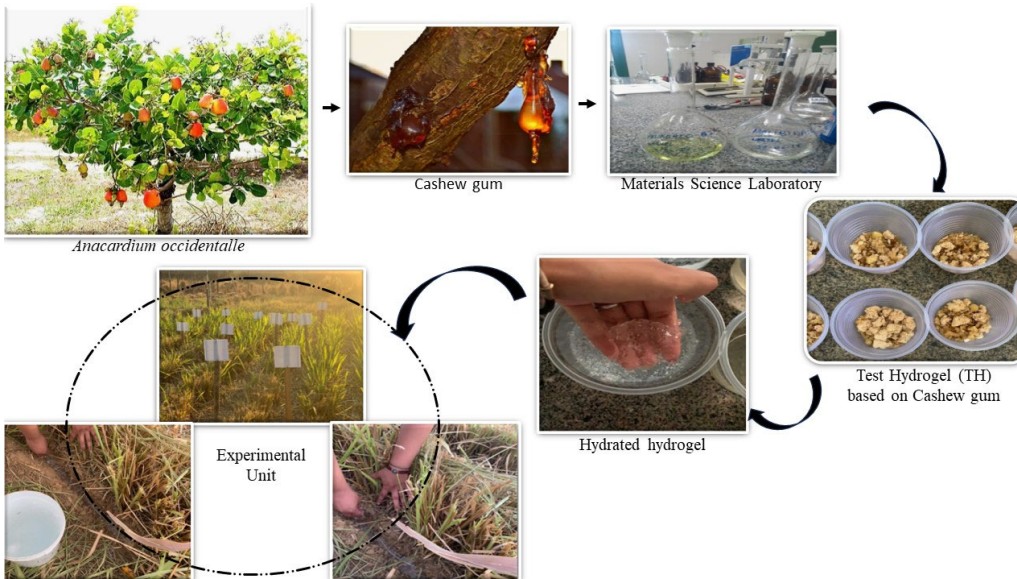

**Figure 1.** Hydrogel produced from cashew exudate and applied to Mombasa grass in the field.

Each gram of TH contained 0.5% $K_3PO_4$, 67% K, and 33% P. The TH had a swelling capacity of 1100 g of $H_2O$ $g^{-1}$ of hydrogel, a thermal stability at 439 °C, and was non-toxic. The TH was produced in the Interdisciplinary Laboratory of Advanced Materials (LIMAV) of the Federal University of Piaui. The test hydrogel was used as recommended according to the characteristics found in the study conducted by [11], who produced hydrogel based on natural fiber from babassu mesocarp (plant scientific name *Orbignya pharelata*), different from the test hydrogel in this experiment, which was produced with fiber obtained from cashew gum (plant scientific name *Anacardium occidentale* L.). In the chemical analysis of the cashew gum hydrogel, the authors concluded that the material demonstrated good swelling capacity in consecutive cycles. The samples exhibited no toxicity, and the results showed are promising for use as a water reserve system and the controlled release of nutrients [12].

The commercial hydrogel used was a copolymer of acrylamide and potassium acrylate branded by Hydroplan-EB (SAP)® as a commercial synthetic product. Following the manufacturer's recommendations, both hydrogels were applied to the soil for planting and previously hydrated with 400 L for each kg of hydrogel.

### 2.4. Experimental Units

The soil was classified as dystrophic red-yellow latosol associated with quartz sands [13], with a sandy loam physical characterization (clay: 220, silt: 50, and sand: 720 g $kg^{-1}$).

After sampling, the soil was dried and sieved through a 2 mm diameter metal mesh to remove impurities and a sample of this soil was taken for physical and chemical analysis following the methodology for determination of the soil chemical characteristics [14], which revealed the following results: 5.2 pH in water; 8.37 mg $dm^{-3}$ phosphorus (P); 0.05 mg $dm^{-3}$ potassium (K); 1.20 cmolc $dm^{-3}$ calcium (Ca); 0.08 cmolc $dm^{-3}$ magnesium (Mg); <0.50 cmolc $dm^{-3}$ aluminum (Al); 2.30 cmolc $dm^{-3}$ hydrogen + aluminum (H + Al); 1.33 cmolc $dm^{-3}$ sum of bases (SB); 3.63 cmolc $dm^{-3}$ CEC at pH 7.0 (T); 36.6% base saturation (V); and 27.3% aluminum saturation (M).

A correction fertilization was conducted in accordance with the recommendations for pasture planting made by [15]. Therefore, as the TH has 67% K and 33% P in its composition, standardized fertilization was performed by applying the equivalent of these minerals in the treatments with no hydrogel or commercial hydrogel.

To correct the potassium levels, potassium chloride (60% $K_2O$) was used as the source of potassium, applying the equivalent of 40 kg ha$^{-1}$ of $K_2O$. Single superphosphate (18% $P_2O_5$) was used as the source of phosphorus, applying the equivalent of 90 kg $P_2O_5$ ha$^{-1}$. In addition, the equivalent of 100 kg of nitrogen ha$^{-1}$ was applied in the form of urea (45% N). The fertilizers were diluted in water and applied separately to the pasture.

The evaluation period lasted for 60 days in a Mombaça grass pasture that had been established for five years, with two cycles of 30 days each to assess growth. At first, the pasture was cut to uniformity, leaving a residual height of 15 cm, and the material was removed from the experimental plots. Subsequently, the commercial hydrogel and the test hydrogel were applied in linear meters in the base of the clumps at an amount of 20 kg ha$^{-1}$.

The application of the hydrogel to the plant was carried out once, after cutting to standardize the plants, in holes 5 cm wide and 5 cm deep, at 10 cm from the plants. The hydrogel was buried after application in the holes; scheme shown in Figure 2.

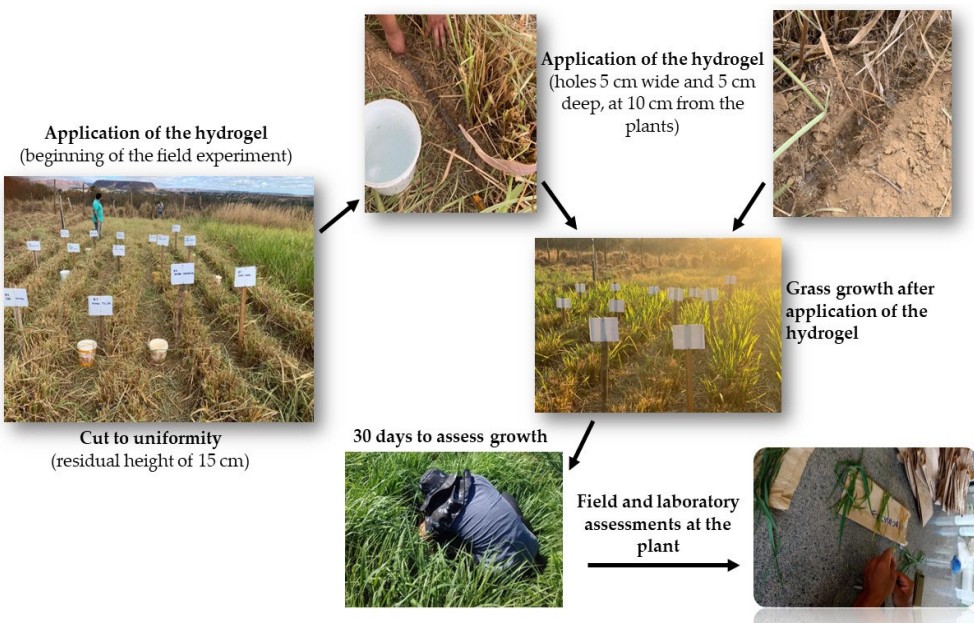

**Figure 2.** Hydrogel applied and evaluated to Mombasa grass in the field.

### 2.5. Evaluation of the Morphogenic and Production Characteristics of the Grasses

Morphogenic characteristics were assessed according to [16], using millimeter-graded rulers every three days during two harvest cycles, with 30 days in each cycle and an experimental period of 60 days. To assess morphogenesis, three tillers were drawn and marked with a smooth plastic-coated wire of different colors. The number of leaves, leaf blade length, stem length, and leaf stage classification (expanding, expanded, senescent, and dead) were monitored on each tiller. With this information, the following parameters were calculated: leaf appearance rate (LAR, leaves tiller$^{-1}$ day$^{-1}$), by dividing the number of leaves emerging per tiller by the number of days in the evaluation period; phylochron (PHY, days leaf$^{-1}$ tiller$^{-1}$), which is the inverse of the leaf appearance rate; leaf elongation rate (LER, cm leaf$^{-1}$ day$^{-1}$), by summing the leaf blade elongation per tiller and dividing by the number of days of evaluation; stem elongation rate (SER, cm tiller$^{-1}$ day$^{-1}$), by summing the stem elongation per tiller and dividing by the number of days of evaluation; number of live expanded leaves per tiller (NLLe), which is the average number of fully expanded leaves per tiller, including sheared leaves; leaf lifespan (LLS, days), which is estimated using the equation LLS (day) = NLLe × PHY [16]; leaf senescence rate (LSR, cm tiller$^{-1}$ day$^{-1}$), by the decrease in length of the green part of the leaf blade, which is obtained from the difference between the initial and final measurements and divided by

the number of days of evaluation; final leaf length (FLL, cm), which is the average length of the live, fully expanded and uncut leaves in the tiller; and final stem length (FSL, cm), which is the average length of the stems.

The appearance rate was evaluated by counting the number of leaves that emerged per tiller, divided by the number of evaluation days (tiller leaves$^{-1}$ day$^{-1}$); the elongation rate was obtained by adding up all the leaf elongation per tiller, divided by the number of days in the evaluation period (tiller mm$^{-1}$ day$^{-1}$); leaf lifespan was obtained by counting the average number of elongating leaves per tiller; phylochron was obtained by the inverse of the rate of appearance of leaves (days); and the number of leaves per tiller was determined by counting the number of live leaves on three tillers in the pasture.

Production characteristics were assessed at the end of each cycle, as follows: plant height, which is measured from the ground to the last expanded leaf of the tallest tiller; and tiller population density (TPD), by counting the number of live tillers present in each clump. After the evaluations, Mombaça grass was cut at a residual height of 15 cm. Green forage samples were taken to the laboratory and weighed on an electronic scale with a precision of 1 g.

Green mass samples were then separated into leaf blades, stems + sheaths, and dead material (DM), then weighed to obtain the leaf green mass yield (LGMY), then placed in identified paper bags and taken to an oven with forced air ventilation for 72 h at 55 °C, until reaching constant weight, in order to obtain the leaf dry mass yield (LDMY), which was converted to kg ha$^{-1}$.

### 2.6. Analysis of the Chemical and Mineral Composition of the Grasses

After pre-drying, samples of the forage dry biomass from each treatment were ground in a stationary "Thomas Wiley" knife-type mill with a 1.0 mm mesh screen for laboratory chemical analysis. The following chemical composition analyses were carried out according to [17]: determination of dry matter (DM) at 105 °C (INCT-CA Method G-003/1), crude protein (CP) (INCT-CA Method N-001/1), mineral matter (MM) (INCT-CA Method M-001/1), neutral detergent fiber (NDF) (INCT-CA Method F-002/1), acid detergent fiber (ADF), and organic matter (OM). All analyses were conducted at the Animal Nutrition Laboratory of UFPI in Bom Jesus/CPCE.

Macro and micronutrient analyses were conducted at the Soil and Plant Analysis Laboratory of the UFPI/Cinobelina Elvas Campus. The following mineral contents were determined: phosphorus (P), potassium (K), calcium (Ca), and magnesium (Mg), expressed in g kg$^{-1}$; iron (Fe), manganese (Mn), zinc (Zn), and copper (Cu), expressed in mg kg$^{-1}$. Nitric–perchloric digestion was conducted, and after digestion, the phosphorus (P) content was determined using UV/VIS spectrophotometry at 660 nm by reading the intensity of the blue color of the phosphomolybdic complex produced by the reduction of molybdate with ascorbic acid in a Digital Light UV-Visible spectrophotometer model IL-592 EVEN. The levels of potassium (K), calcium (Ca), magnesium (Mg), and zinc (Zn) were determined using atomic absorption spectrophotometry (AAS), model AA240FS VA-RIAN$^{®}$. The methodology used to analyze the plant nutrients followed the standards described in [18].

### 2.7. Statistical Analysis

The data were subjected to analysis of variance of the main effects and the harvest cycles × hydrogel interaction. When the interaction effect was significant, the factors were split using the Scott–Knott test to compare the means, considering a 5% significance level ($p \leq 0.05$). Statistical analyses were conducted using SISVAR software version 5.0 [19].

## 3. Results

### 3.1. Morphogenesis

No effect of treatment ($p > 0.05$) was found on SER, LSR, NLLe, LAR, and PHY (Table 1). On the other hand, LAR was higher ($p < 0.05$) when Mombaça grass was hydrated

with test hydrogel (42.33 mm day$^{-1}$), which was different from the no-hydrogel treatment (30.53 mm day$^{-1}$).

**Table 1.** Morphogenic characteristics of Mombaça grass pasture under three forms of hydration.

| Variables | Hydrogel (H) | | | Mean | *p*-Value | SEM |
|---|---|---|---|---|---|---|
| | No | Commercial | Test | | | |
| LER (mm tiller$^{-1}$ day$^{-1}$) | 30.53 b | 35.00 ab | 42.33 a | 35.95 | 0.03 | 2.67 |
| SER (mm tiller$^{-1}$ day$^{-1}$) | 0.26 a | 0.35 a | 0.60 a | 0.40 | 0.11 | 0.10 |
| LSR (mm tiller$^{-1}$ day$^{-1}$) | 0.95 a | 1.01 a | 1.36 a | 3.36 | 0.36 | 4.22 |
| NLLe (leaves tiller$^{-1}$) | 8.27 a | 8.53 a | 9.33 a | 8.71 | 0.87 | 1.47 |
| LAR (leaves tiller$^{-1}$ day$^{-1}$) | 0.257 a | 0.360 a | 0.495 a | 0.371 | 0.53 | 0.144 |
| PHY (days leaves$^{-1}$ tiller$^{-1}$) | 2.86 a | 4.55 a | 4.97 a | 4.13 | 0.51 | 1.32 |

LER: Leaf Elongation Rate; SER: Stem Elongation Rate; LSR: Leaf Senescence Rate; NLLe: Number of Expanded Live Leaves; LAR: Leaf Appearance Rate; PHY: Phylochron. Different lowercase letters in the row represent the significance of $p < 0.05$ for the use of hydrogel (H). SEM: Standard Error of the Mean.

Variables SER, LAR, and PHY showed 0.60 mm tiller$^{-1}$ day$^{-1}$, 0.495 leaves tiller$^{-1}$ day$^{-1}$, and 4.97 days leaves$^{-1}$ tiller$^{-1}$, respectively, in a treatment TH, while in a treatment NH, they showed 0.26 mm tiller$^{-1}$ day$^{-1}$, 0.257 leaves tiller$^{-1}$ day$^{-1}$, and 2.86 days leaves$^{-1}$ tiller$^{-1}$. LSR and NLLe showed averages of 3.36 mm tiller$^{-1}$ day$^{-1}$ and 8.71 leaves tiller$^{-1}$.

### 3.2. Structure and Production Characteristics

There was no significant difference between the heights of Mombaça grass, which ranged from 59 to 64.8 cm. On the other hand, there was an isolated effect ($p < 0.05$) between hydration forms and harvesting cycles on LGMY. There was also no effect of interaction ($p < 0.05$) between the forms of hydration and harvesting cycles on LDMY. It was found that there was an isolated effect between the forms of hydration on TPD, with a difference in the superiority of the test hydrogel when compared to the management without hydrogel, Table 2.

**Table 2.** Structural and production characteristics of Mombaça grass pasture under three forms of hydration.

| Hydrogel | Cycles | | Mean | SEM | *p*-Value | | |
|---|---|---|---|---|---|---|---|
| | Cycle 1 | Cycle 2 | | | H | C | H × C |
| | Plant Height (cm) | | | | | | |
| No | 64.8 | 64.2 | 64.5 | 1.32 | 0.23 | 0.62 | 0.29 |
| Commercial | 64.6 | 59.0 | 61.8 | | | | |
| Test | 62.2 | 63.8 | 63.0 | | | | |
| Mean | 63.8 | 62.3 | | | | | |
| | Leaf Green Mass Yield (kg ha$^{-1}$) | | | | | | |
| No | 7000.0 | 4936.0 | 5968.0 B | 232.17 | <0.01 | <0.01 | 0.84 |
| Commercial | 7448.0 | 5296.0 | 6372.0 B | | | | |
| Test | 8664.0 | 6161.0 | 7412.0 A | | | | |
| Mean | 7704.0 a | 5464.0 b | | | | | |
| | Leaf Dry Mass Yield (kg ha$^{-1}$) | | | | | | |
| No | 2617.4 Ca | 1678.8 Ab | 2338.9 | 81.55 | 0.16 | <0.01 | <0.01 |
| Commercial | 3694.5 Ba | 1966.5 Ab | 2830.5 | | | | |
| Test | 4477.0 Aa | 2060.4 Ab | 3077.9 | | | | |
| Mean | 3596.3 | 1901.9 | | | | | |

**Table 2.** *Cont.*

| Hydrogel | Cycles | | Mean | SEM | p-Value | | |
| --- | --- | --- | --- | --- | --- | --- | --- |
| | Cycle 1 | Cycle 2 | | | H | C | H × C |
| Tiller Population Density (TPD, tiller m$^{-2}$) | | | | | | | |
| No | 177.0 | 188.2 | 182.6 b | 9.76 | <0.01 | 0.09 | 0.44 |
| Commercial | 169.6 | 219.4 | 194.5 ab | | | | |
| Test | 231.0 | 243.4 | 237.3 a | | | | |
| Mean | 192.6 a | 217.0 a | | | | | |

Different lowercase letters in the row represent the significance of $p < 0.05$ for the use of hydrogel (H). Different uppercase letters in the column represent the significance of $p < 0.05$ for the harvesting cycle (C). SEM: Standard Error of the Mean.

### 3.3. Chemical and Mineral Composition

The use of hydrogels did not affect the chemical composition of Mombaça grass. The average dry matter content was 308.6 g kg$^{-1}$, crude protein was 71.8 g kg$^{-1}$ DM, neutral detergent fiber was 707.6 g kg$^{-1}$ DM, acid detergent fiber showed 347.9 g kg$^{-1}$ DM, and mineral matter showed 43.3 g kg$^{-1}$ DM; Table 3.

**Table 3.** Chemical composition of Mombaça grass pasture under three forms of hydration.

| Variables | Hydrogel (H) | | | Mean | SEM | p-Value |
| --- | --- | --- | --- | --- | --- | --- |
| | No | Commercial | Test | | | |
| DM (g kg) | 310.6 a | 307.1 a | 308.1 a | 308.6 | 1.53 | 0.98 |
| CP (g kg$^{-1}$ DM) | 69.5 a | 73.8 a | 72.3 a | 71.8 | 0.35 | 0.68 |
| NDF (g kg$^{-1}$ DM) | 713.6 a | 691.5 a | 717.9 a | 707.6 | 0.83 | 0.11 |
| ADF (g kg$^{-1}$ DM) | 349.3 a | 341.2 a | 353.4 a | 347.9 | 0.57 | 0.36 |
| MM (g kg$^{-1}$ DM) | 42.2 a | 41.9 a | 46.0 a | 43.3 | 0.44 | 0.77 |

DM: Dry Matter; CP: Crude Protein; NDF: Neutral Detergent Fiber; ADF: Acid Detergent Fiber; MM: Mineral Matter. SEM: Standard Error of the Mean. Different lowercase letters in the row represent the significance of $p < 0.05$ for the use of hydrogel (H).

In evaluating the mineral composition of the Mombaça grass under different hydration forms, there was an effect ($p < 0.05$) only on Zn, with an accumulation of 28.0 mg kg$^{-1}$ for the hydration with the TH. The macronutrients Mg, P, and K showed averages of 4.5 g kg$^{-1}$, 4.0 g kg$^{-1}$, and 331.0 g kg$^{-1}$, respectively, while Cu showed 8.1 mg kg$^{-1}$, Mn 124.6 mg kg$^{-1}$, Fe 78.6 mg kg$^{-1}$, and Zn 13.6 mg kg$^{-1}$; Table 4.

**Table 4.** Mineral composition of Mombaça grass pasture under three forms of hydration.

| Variables | Hydrogel (H) | | | Mean | SEM | p-Value |
| --- | --- | --- | --- | --- | --- | --- |
| | No | Commercial | Test | | | |
| Ca (mg kg$^{-1}$) | 42.6 a | 35.7 a | 35.2 a | 37.8 | 0.31 | 0.24 |
| Mg (mg kg$^{-1}$) | 5.0 a | 4.5 a | 4.1 a | 4.5 | 0.07 | 0.72 |
| P (mg kg$^{-1}$) | 3.5 a | 3.8 a | 4.7 a | 4.0 | 0.05 | 0.35 |
| K (mg kg$^{-1}$) | 365.2 a | 334.0 a | 294.0 a | 331.0 | 4.07 | 0.49 |
| Cu (mg kg$^{-1}$) | 1.1 a | 1.4 a | 17.0 a | 8.1 | 0.03 | 0.48 |
| Mn (mg kg$^{-1}$) | 35.9 a | 34.1 a | 304.0 a | 124.6 | 0.59 | 0.80 |
| Fe (mg kg$^{-1}$) | 19.3 a | 23.5 a | 193.0 a | 78.6 | 0.37 | 0.72 |
| Zn (mg kg$^{-1}$) | 4.6 b | 8.3 b | 28.0 a | 13.6 | 0.11 | 0.02 |

Ca: Calcium; Mg: Magnesium; P: Phosphorus; K: Potassium; Cu: Copper; Mn: Manganese; Fe: Iron; Zn: Zinc. SEM: Standard Error of the Mean. Different lowercase letters in the row represent the significance of $p < 0.05$ for the use of hydrogel (H).

## 4. Discussion

### 4.1. Morphogenesis

The results from the pasture under different forms of hydration only showed an effect on LER from the use of the test hydrogel, which presented 42.33 mm leaf$^{-1}$ day$^{-1}$, being superior to the no-hydrogel management but statistically similar to the commercial hydrogel treatment, which is a synthetic product with the capacity to absorb 400 times its weight [12,20]. Therefore, this result shows that CH can be safely replaced by the TH, as it is an organic and 100% biodegradable product with high water retention capacity [21]. In addition, CH was similar to NH management and did not influence any of the morphogenic variables evaluated.

However, there were no differences between hydration forms used in the Mombaça grass pasture for SER, LSR, NLLe, and PHY, although these variables showed higher average values in the TH treatment. It can, therefore, be seen that TH favors the appearance and expansion of new leaves and may influence the other variables over the course of the harvest cycles, as the main characteristic of hydrogel is that it releases water gradually [22,23]. This can be seen in the LAR and NLLe values, which practically doubled compared to the NH management.

Evaluations of morphogenic characteristics [24], structural characteristics [25], and production characteristics associated with the evaluation of the plant's chemical and mineral composition are important tools that indicate the desirable production indices in the plants, and with this, it is possible to define the appropriate management and crop treatments for the plants [26,27].

### 4.2. Structure and Production Characteristics

It was observed that hydration with the test hydrogel and the commercial hydrogel had different effects on LGMY. The TH is favorable to the production of Mombaça grass and was superior to no hydrogel for the same variable. Comparing the harvest cycles, it was observed that the second cycle had lower LGMY. This was because the water in the hydrogel was released in greater quantities in the first 30 days (the first cycle) [21], enhancing leaf elongation and the growth of the Mombaça grass. Therefore, more water available in the pasture led to greater LGMY, which was observed in the morphogenic characteristics, in which LER performed better and is directly related to leaf growth and production.

An effect was observed in the interaction between cycles and forms of hydration on LDMY in the first harvesting cycle, which means that the first cycle had higher LDMY than cycle two. However, Mombaça grass had higher LDMY when using the TH in both the first and second cycles. LDMY was lower when the hydrogels were not used.

TPD was higher in the TH treatment in comparison to NH. This suggests that the continued use of hydrogels will result in a stable LGMY since the more tillers appear, the greater the grass production will be [28]. Also, more harvesting cycles are needed to stabilize TPD in the pasture under the effect of CH and the TH, showing different responses to TPD between these hydrogels, as it is a very unstable variable. The advantage of using the test hydrogel, according to [12], is because the hydrogel does not demonstrate toxicity and has the potential for application in agriculture.

### 4.3. Chemical and Mineral Composition

The lack of effect in the chemical composition analysis between the forms of hydration used in Mombaça grass pasture shows that hydrogels do not interfere with its composition, although changes may occur in the case of evaluations with other forage grasses. Although statistically, the values between treatments were similar, there was an improvement in the levels of CP, NDF, ADF, and MM with the test hydrogel. NDF values were within the expected range for this forage species subjected to a 30-day rest period and residual harvesting height of 15 cm [28,29]. For the other variables, the increase under the effect of the TH is positive for the species. The use of a hydrogel based on the natural fiber of *Orbignya phalerata* provided a better result in terms of the chemical composition of the

cactus pear plant when compared to the use of commercial hydrogel and the non-use of hydrogel; similar effects were observed by [6] in their study.

The increase in the levels of micronutrients evaluated may have been due to the fact that they are easy to mobilize through mass flow in the direction of the water released by the hydrogel [30,31]. Zn was the only chemical element that had a significant effect on the three forms of hydration. The Zn content observed was 28 mg kg$^{-1}$ in the treatment TH, despite this micronutrient showing antagonistic absorption to phosphorus [32,33]. In addition, critical toxic levels of Zn in grasses occur when there is more than 110 mg Zn kg$^{-1}$ of soil [34]. The contents of macronutrients Ca, Mg, and K decreased with the use of the TH. For the concentration of Ca in Poaceae species, the maximum value found was 4.8 g kg$^{-1}$, which corresponds to 0.48% in the *Eleusine indicae* species and a minimum of 0.6 g kg$^{-1}$ equivalent to 0.06% in the *Aristida longifolia* species [33]. Potassium does not play a structural role in the plant, but it does contribute to its osmotic regulation. Under conditions of deficiency, potassium can move from the maturing organs to those that are growing. In the case of potassium deficiency in plants, the appropriate level for this nutrient is 10.0 to 14.0 g kg$^{-1}$ [32]. As the values vary according to the organ analyzed, the species analyzed in this study have a high concentration of this nutrient, according to the results.

The literature reports that phosphorus is one of the macronutrients that are in little demand in pasture for optimal growth of forage plants, and it varies depending on the forage species [15]. P can account for between 0.1 and 0.5% of dry matter, and the appropriate levels of this nutrient in Mombaça grass are between 1.0 and 2.3 g kg$^{-1}$. The P content observed in this study is within these limits. The authors of [32] establish values of 0.11 to 0.30 g kg$^{-1}$. Low phosphorus concentrations in the cell cytoplasm reduce the activity of cytoplasmic proteins and minimize forage plant growth. When growth is paralyzed, the older leaves of the plants are deficient in phosphorus, and at first show a dark green and bluish staining caused by the higher relative concentration of chlorophyll, and purple tones can occur in the leaves and stem [32,33].

## 5. Conclusions

The cashew gum hydrogel increased the leaf elongation rate to 42.33 mm tiller$^{-1}$ day$^{-1}$ and green leaf production to 8664.0 kg ha$^{-1}$, in addition to stimulated tillering.

It increased the amount of Zn with an accumulation of 28.0 mg/kg in Mombaça grass. However, there was no effect of hydration on the chemical composition.

The biodegradable hydrogel has the potential to be used in regions of tropical climate and dystrophic red-yellow latosols and could replace commercial synthetic hydrogels.

It is recommended to use the hydrogel produced from cashew gum in Mombasa grass pastures. It is also recommended that further studies be carried out with this hydrogel on other grass species over the long term so that a more reliable recommendation can be made.

**Author Contributions:** Methodology, D.M.A.B., L.F.R.C., R.R.d.N. and J.P.M.P.; formal analysis, R.L.E. and L.R.B.; resources, E.C.S.-F.; writing—original draft preparation, D.M.A.B. and M.J.d.A.; writing—review and editing, H.R.d.S.; supervision, R.L.E.; project administration, R.L.E., L.R.B. and E.C.S.-F.; funding acquisition, E.C.S.-F. All authors have read and agreed to the published version of the manuscript.

**Funding:** This work was partially supported by the Brazilian agencies MCTIC/CNPq (Grant n° 406973/2022-9) through INCT/Polysaccharides (National Technology-Science Institute for Polysaccharides).

**Institutional Review Board Statement:** Not applicable.

**Data Availability Statement:** The data presented in this study are available upon request from the corresponding author.

**Conflicts of Interest:** The authors declare no conflict of interest.

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
