# Peer review of "Hydrogel Based on Cashew Gum and Polyacrylamide as a Potential Water Supplier in Mombaça Grass Pastures: A Sustainable Alternative for Agriculture"

_sustainability, doi:10.3390/su152316423_

Round 1

Reviewer 1 Report

Comments and Suggestions for Authors

This study compared biodegradable cashew gum hydrogels with synthetic hydrogels in Mombaça grass pastures. The biodegradable hydrogel improved leaf growth, green leaf production, and tillering. It also positively affected the mineral composition of the grass. The research suggests that these eco-friendly hydrogels can replace synthetic ones for better pasture performance.

Some constructive comments:

1. The experiment in this manuscript compares the performance of the Synthetic commercial hydrogel (CH), made from a synthetic polyacrylamide product, with the Biodegradable test hydrogel (TH) in Mombaça grass pastures. However, the manuscript could benefit from a more explicit explanation of why Mombaça grass pastures show improved growth with TH compared to CH.

2. The absence of figures in the manuscript limits the visual representation of the experimental setup and data interpretation. Including figures would greatly enhance the reader's ability to understand and compare the results.

3. The excessive use of abbreviations in tables and the main text can impede readability and comprehension. Consider using the full names of terms when they are not essential as abbreviations, to enhance clarity and readability.

4. Consider enhancing the introduction by describing the hydrogels employed in previous research, along with any documented results or benefits. This added context will provide readers with a deeper understanding of the background and context of your study. Additionally, it would be valuable to clarify the innovative aspects of your research in comparison to prior studies.

Comments on the Quality of English Language

The quality of the English language in this manuscript is good and reads fluently. The writing is clear and easily understandable except some abbreviations.

Author Response

The corrections are attached in the file below.

Reviewer 2 Report

Comments and Suggestions for Authors

The research manuscript " Biodegradable cashew gum hydrogel in Mombasa grass pasture: a sustainable option for agriculture" targeted the agricultural field. Some ambiguities should be considered.

In this study, authors evaluated the capability of replacing synthetic hydrogels derived from petroleum with biodegradable hydrogels in Mombaça grass pastures. Their results showed that compared with No hydrogel and Synthetic commercial hydrogel, cashew gum hydrogels had remarkable effects in increasing leaf elongation rate, green leaf production, and stimulating tillering.

 Please answer the following comments in detail:

1.      Please delete the point (.) from the end of the title.

2.      It is advisable to consider rewriting the abstract. It is preferable for the "aim of study" to be included towards the conclusion of the abstract.

3.      The introduction is too short. It is recommended that the authors discuss the application potential of different biopolymers as hydrogels in agriculture. There are several types of biopolymers, such as alginate and CMC, for potentially preparing hydrogel. Then it is recommended that authors discuss the place of gums in preparing hydrogels.

In this context, the authors should be cited properly ref such as DOI: 10.3390/polym15020440; DOI: 10.3390/mi13091423

4.      Nowadays, plants are worthwhile bioresources for the extraction of gums, such as Pistacia atlantica, Amygdalus scoparia, … . The authors should explain why they preferred cashew gum for preparing hydrogels.

5.      If possible, it is recommended that the authors place a figure from the morphology and texture of all prepared hydrogel in the result.

Author Response

The corrections are attached in the file below.
Please see the attachment.

Reviewer 3 Report

Comments and Suggestions for Authors

The presented manuscript “Biodegradable cashew gum hydrogel in Mombasa grass pasture: a sustainable option for agriculture” has been received for review. The presented study is an interesting study describing the use of the biodegradable hydrogel in agriculture. The manuscript is of importance to the journal and it is written in good grammar and syntax. It can be considered for publication after incorporating following suggestions:

1.      The title does not describe precisely the work performed in the study. It should be modified in such as way that it describes the major application of the study.

2.      Keywords: chemical composition is not effective. They should be removed.

3.      Introduction: The introduction is too short and does not give sufficient information about th study. It should include background, review of literature containing recent studies performed related to the study, method, aim and scope of the study. Only two reports have been added.

4.      Conclusion has not been written properly. It should include a detailed summary from each result section, overall conclusion from the study and future recommendations should be suggested.

5.      This study was conducted in particular geographic location. Conclusion should be added with the information about the hydrogel applicability in different geographic locations such as countries in different continents based on their geographical conditions. It will give more novelty towards the application of the material produced.

6.      What is morphogenesis? It should be explained somewhere in the manuscript to make it clear to the reader.

Author Response

(The authors gave the same response as above.)

Reviewer 4 Report

Comments and Suggestions for Authors

The authors reported in this manuscript “Biodegradable cashew gum hydrogel in Mombasa grass pasture: a sustainable option for agriculture” (sustainability-2667486) the use of synthetic hydrogels derived from petroleum by biodegradable hydrogels in Mombaça grass pastures (Megathyrsus maximum. Although the authors have worked on an interesting area of research, however, their explanation is unclear to understand the strategy and experiments. This reviewer feels that the manuscript is incomplete and insufficient to publish in the Journal. There are crucial lacks in the manuscript. 

1. In general, there are many grammatical and typographical errors and many sentences are inconsistent and incomprehensible and need to be revised for improve logical flow and clarity; the authors have to check the whole manuscript carefully to eliminate all types of errors.

To name few; 

Title; “agriculture.” should be “agriculture”

Abstract, L 2; “by” should be “with”

Abstract, L 3; “(Megathyrsus maximum)” should be ““(Megathyrsus maximus)”

Abstract, L 4; “No hydrogel” should be “no hydrogel”

Abstract, L 4; “Synthetic commercial” should be “synthetic commercial”

Abstract, L 5; “Biodegradable test” should be “biodegradable test”

Abstract, L 6; “experimental design was” should be “experimental design consisted of”

……and so on

A complete proofreading is needed.

2. In general, please check the definition`s articles throughout the manuscript.

3. In introduction part; it is very poor in up to date citation regarding physicochemical features and biological characteristics of natural gum hydrogels as well as their applications.

4. In the whole of the manuscript, the explanation of experimental design is unclear, insufficient, and not supported with relevant images showing different stages of plant treatment.

5. Section 2.3 ‟Synthesis of Hydrogels used”: this part is vague and incomprehensible and should be rewritten in more detail

6. Their is no results and discutuions 

7. There are not enough measurements and information about the physicochemical and structrual characterization of new hydrogels (FTIR, UV, elemental analysis, XPS, EDX, zeta potential). These tools are very important to validate their successful formation.  

8. As the loading capacity (LC) encapsulation efficiency (EE) of hydrogel are very important parameters for any applications; thus these parameters should be investigated by the authors.  

9. Lack of comparison of the findings of this study with those previously reported in the literature.

Comments on the Quality of English Language

Extensive editing of English language required

Author Response

(The authors gave the same response as above.)

Round 2

Reviewer 3 Report

Comments and Suggestions for Authors

The manuscript has been sufficiently improved.

Author Response

Thank you.

Reviewer 4 Report

Comments and Suggestions for Authors

The manuscript has undergone partial revisions, yet it still exhibits significant shortcomings, rendering it unsuitable for publication at this stage. Here are the specific issues that need to be addressed:

1. Still, there are not enough measurements and information about the physicochemical and structural characterization of new hydrogels (FTIR, UV, elemental analysis, XPS, EDX, zeta potential). These tools are very important to validate their successful formation..

2. It is imperative to include the loading capacity (LC) encapsulation efficiency (EE) of the hydrogel.

3. Still, the experimental design is unclear, insufficient, and not supported with relevant images showing different stages of plant treatment..

Comments on the Quality of English Language

 Minor editing of English language required

Author Response

Thank you.